# Unified DeepLabV3+ for Semi-Dark Image Semantic Segmentation

**DOI:** 10.3390/s22145312

**Published:** 2022-07-15

**Authors:** Mehak Maqbool Memon, Manzoor Ahmed Hashmani, Aisha Zahid Junejo, Syed Sajjad Rizvi, Kamran Raza

**Affiliations:** 1High Performance Cloud Computing Center (HPC3), Department of Computer and Information Sciences, Universiti Teknologi PETRONAS, Seri Iskandar 32610, Malaysia; mehak_19001057@utp.edu.my (M.M.M.); aisha_19001022@utp.edu.my (A.Z.J.); 2Department of Computer Science, Shaheed Zulfiqar Ali Bhutto Institute of Science and Technology, Karachi 75600, Pakistan; sshussainr@gmail.com; 3Faculty of Engineering Science and Technology, Iqra University, Karachi 75600, Pakistan; kraza@iqra.edu.pk

**Keywords:** semantic segmentation, super-pixels, atrous convolutions, high-resolution images, urban environments

## Abstract

Semantic segmentation for accurate visual perception is a critical task in computer vision. In principle, the automatic classification of dynamic visual scenes using predefined object classes remains unresolved. The challenging problems of learning deep convolution neural networks, specifically ResNet-based DeepLabV3+ (the most recent version), are threefold. The problems arise due to (1) biased centric exploitations of filter masks, (2) lower representational power of residual networks due to identity shortcuts, and (3) a loss of spatial relationship by using per-pixel primitives. To solve these problems, we present a proficient approach based on DeepLabV3+, along with an added evaluation metric, namely, Unified DeepLabV3+ and $S^{3}core$, respectively. The presented unified version reduced the effect of biased exploitations via additional dilated convolution layers with customized dilation rates. We further tackled the problem of representational power by introducing non-linear group normalization shortcuts to solve the focused problem of semi-dark images. Meanwhile, to keep track of the spatial relationships in terms of the global and local contexts, geometrically bunched pixel cues were used. We accumulated all the proposed variants of DeepLabV3+ to propose Unified DeepLabV3+ for accurate visual decisions. Finally, the proposed $S^{3}core$ evaluation metric was based on the weighted combination of three different accuracy measures, i.e., the pixel accuracy, IoU (intersection over union), and Mean BFScore, as robust identification criteria. Extensive experimental analysis performed over a CamVid dataset confirmed the applicability of the proposed solution for autonomous vehicles and robotics for outdoor settings. The experimental analysis showed that the proposed Unified DeepLabV3+ outperformed DeepLabV3+ by a margin of 3% in terms of the class-wise pixel accuracy, along with a higher $S^{3}core$, depicting the effectiveness of the proposed approach.

## 1. Introduction

The scene comprehension of urban environments is the most crucial component of the autonomous vehicle industry. Since the comprehension and understanding of visual environments include dynamic visual environments, the deployment of scene comprehension demands the optimal discovery of all the objects present in the incoming visual scene, even if the incoming image is captured in low-light conditions. The rapid advancements in the domain of the deep learning (DL) paradigm illustrate promising solutions that focus on different problem domains. The methods proposed in the DL paradigm present satisfactory performance for the classification, detection, and segmentation of normal (bright) images. However, these methods display performance degradation when it comes to semi-dark imagery [1,2]. Furthermore, for visual image comprehension in an urban setting, the automated solution will face diversity in terms of lightning conditions. For scene comprehension in autonomous vehicles, a widely used alternative is semantic segmentation. Semantic segmentation results in a pixel-wise understanding of a visual image. Each pixel of an image receives an object label, which then helps to create an object mask that presents the global context of the image, and hence, the autonomous application (vehicle or robot) can make an informed decision. Recent research showed that DL-based semantic segmentations can be successfully employed for remote sensing imagery, including radar images [3] and high-resolution (HR) images from indoor, outdoor, agriculture, and industrial settings [4,5,6]. By leveraging the advancements of deep neural networks (DNNs) in the DL paradigm, more focused solutions can be proposed to deal with the diversity (including semi-dark imagery) involved in visual scene understanding. For the sematic segmentation of semi-dark imagery, dark pixel values pose great challenges to DNNs for the classification of HR images. Traditional fully convolution neural networks identify the object class to a certain extent but cannot handle variable image sizes and suffer from the retainment of the spatial component of a classified pixel [7]. To deal with these problems, a different architectural mechanism was presented, namely, an encoder–decoder approach [8]. This approach tackles problems associated with earlier CNNs that use an encoder, which encapsulates the image information by extracting image features. This feature extraction is accomplished by using any pretrained network. The encoder is then followed by a decoder, which projects the discriminative features (with a low resolution) coming from the encoder over a high-resolution pixel mask [9]. Over time, a lot of different variations of the encoder–decoder architecture were presented [10,11,12]. Encoder–decoders were widely used for automated scene comprehension. Specifically, the DeepLab family has evolved rapidly and has made innovative achievements [10,13,14]. However, even with the recent developments of DeepLab, the optimal semantic segmentation of semi-dark images remains an open area of research. The functionality of the state-of-the-art semantic segmentation modules is constrained by several factors. First, restricted multiscale feature extraction is witnessed due to the usage of biased centric exploitations of the receptive field. The receptive field is the portion of the image that is under consideration by the DCNN for the feature extraction. Centric exploitation results in a greater weightage being assigned to the center pixels and less weightage assigned to the corner pixels for the calculation of output features. This implies that the multiscale information residing in the corners of the receptive field is likely to be skipped by the network. Furthermore, the pretrained network used for an encoder, such as ResNet [12], results in reduced representational power and suffers from network training problems. These problems are attributed to the presence of identity shortcuts in the DCNN and the occurrence of an internal covariate shift during the training phase [15]. Finally, the loss of spatial relationships is also observed as the pixels are classified at the pixel level by using per-pixel primitives, due to which the relationship between the global and local context of the image is lost. Hence, all these problems contribute to inaccurate semantic segmentation of semi-dark imagery. In summary, the existing DL pool of knowledge is limited in the following aspects regarding the semantic segmentation of semi-dark imagery in outdoor settings:The loss of spatial relationships by using per-pixel primitives, resulting in poorly detected object classes over boundary pixels.The limited representational power due to the use of identity shortcuts and the occurrence of an internal covariant shift during the training of DNNs.The constrained multiscale feature extraction caused by biased centric exploitations.

Although the encoder–decoder architectures have brought promising progress, the existing state-of-the-art methods fail to produce exact pixel-wise classification in semi-dark image scenarios. All the above-mentioned problems contribute to an inherent loss of information. On the other hand, the increasing demand for optimal automated solutions points to the critical need for generalized solutions in terms of lightning conditions (dark, semi-dark, and bright). To overcome the effect of all the stated problems regarding semi-dark images, we proposed a combined approach using ML-based bunched pixel primitives with enhanced DeepLabV3+ for semantic segmentation. Alongside this, we also proposed an accurate representative measure for accuracy computation. Overall, the presented research produced the following contributions:*1*.*Proposal of a novel version of DeepLabV3+ for semi-dark imagery—Unified DeepLabV3+*

An enhanced version of DeepLabV3+ that provides a unified solution for handling the challenges associated with the comprehension of semi-dark imagery is presented.

*Geometrical pixel abstraction as the input:* To keep the spatial relationship intact, the image is first passed through a machine learning (ML)-based preprocessor to highlight the local pixel structure based on the color and proximity. This generates a high level of pixel abstraction to retain the local spatial information, which in turn helps to map the local context (local pixel structure) to the global context of the image to correctly identify the object class in the final semantic segmentation map.

*Encoder enhancement:* To increase the representational power of the base DeepLab version, the ResNet encoder was updated. The enhancement was focused on boosting the network’s feature extraction by replacing identity shortcuts with non-linear (ReLU) shortcuts. Moreover, to deal with the network training problem, grouped normalization layers are followed by non-linear shortcuts. The usage of group normalization provides better generalization, solving the problem of the internal covariate shift that occurs during the network-training phase.

*Decoder enhancement:* To induce neutrality for all the pixels of the receptive field, dilated convolution layers are usually stacked, which reduces the effect of centric exploitations. Dilated convolutions are also termed atrous spatial pyramid pooling in DeepLabV3+. Following the same idea of pyramid pooling by means of hole (zero) insertion in the convolution filter, we present customized dilation rates (which decide the rate/number of zeros) for semi-dark images. The customized dilation rates are carefully chosen based on the rationale of extracting as much information as possible for optimal semantic segmentation in terms of the resulting class-wise accuracy.

*2*.
*Proposal of a novel evaluation metric—*

$S^{3}core$



The existing evaluation metric relating to the accurate identification of network semantic segmentation does not provide a detailed understanding of the presented visual scene. The state-of-the-art evaluation metric presents incomplete representation since it produces inaccurate pixel class prediction. Thus, we present a balanced evaluation metric that can be used as a representative metric for semantic segmentation, providing a robust evaluation for all scenarios.

The following sections of this article are organized as follows: Section 2 introduces the feature-encoder-based methods, the increased resolution of the feature-encoder-based methods closely related to the presented work, and these methods’ relevance to the applicability for semi-dark imagery. Then, we comprehensively describe our proposed encoder–decoder architecture in Section 3. In Section 4, we present the experimental setup, evaluation metrics, and the discussion of the creation of robust evaluation criteria. Same section presents the results of extensive experimental analysis and comparison with other state-of-the-art methods. Section 5 discusses the major findings and pertinence of the presented research for real-world applications. Finally, Section 6 concludes the research and points out the future directions.

## 2. Related Work

Semantic segmentation is also known as a dense prediction. Semantic segmentation is considered a dense prediction because all the pixels of the image obtain a certain level that creates a resultant pixel mask that highlights the precise object boundaries. The difference between semantic segmentation and instance segmentation is the addition of object semantics in terms of object labels (car, tree, bus, etc.) and object instance semantics in terms of the occurrence of a certain object (car1, car2, …, bus1, bus2, …, etc.) [8]. Figure 1 shows an image and its subsequent semantic segmentation.

Figure 1 shows that the prediction labels are inferred for each pixel, hence this makes it a dense prediction. For semantic segmentation, different methods are available in the DL domain. We focused on two classes of feature encoders, both of which show outstanding progress with comprehensive results and ease of applicability, as well as increasing the resolution of feature-encoder-based methods.

### 2.1. Feature-Encoder-Based Methods

The widely used DL networks that implement feature-encoder-based architectures are VGG [9] and ResNet [12]. The base mechanism employs stacked convolution layers, ReLU, and pooling layers to extract features from the image. The currently used networks take inspiration from these base state-of-the-art encoder networks for presenting their variants by performing transfer learning or training their networks with additional architectural changes.

VGG contributes to a network architecture by incrementing the convolutional layers, which increases the depth of a network. VGG can take only fixed-sized images due to the presence of fully connected layers. Moreover, it uses the subtraction of RGB values and suffers from centric exploitations. Since all the pixel values in semi-dark scenarios reach maximum values, the mean calculation ends up with a loss of information by providing the same numeric value for the local pixels. Furthermore, the implemented design rules preserve complexity per layer [16]. In turn, all these constraints result in the poor localization of objects in terms of misclassified boundary pixels. ResNet contributes in terms of depth and identity connections from initial network layers to the later layer to reduce the spatial localization problem of the object. However, the invariance of the spatial transformation still leads to a loss of fine details of the visual scene. ResNet presents some other challenges, including limited representational power due to the usage of identity shortcuts and the occurrence of an internal covariant shift [15,17]. These challenges further increase the possibility of lower network performance for the focused image problem. Moreover, neither mentioned network was tested for semi-dark images exclusively, implying worse semantic segmentation results for such visual scenarios.

### 2.2. Increased Resolution of Feature-Encoder-Based Methods

To deal with the problems of feature encoder methods, increased resolution of feature-encoder-based methods has been proposed in the literature. The enhancement is suggested to enhance the spatial resolution by incorporating different ML and DL concepts.

First, enhancement is used as a step to deal with the problem associated with the field of view (FoV) to obtain high-resolution maps. To increase the FoV and to deal with the problem of centric exploitations, ‘Atrous convolutions’ are used. An atrous convolution involves the addition of holes/zeros in the convolution filter. Since there are holes in the filter, the focus is diverted from the center pixels to there being equal weightage for all the pixels. The key is to stack atrous convolution layers and not to increase the network’s memory strain. This concept is widely used in the DeepLab network family (further critically analyzed in Section 2.3), where the DeepLab networks use the concept of an encoder–decoder architecture. The encoder takes the input image, generates a high-dimensional feature vector, and aggregates them at multiple levels. The decoder network generates a semantic segmentation mask by upsampling the feature vector generated by the encoder. For the encoder, the DeepLab networks have used different existing networks, including VGG and ResNet. The best performance was obtained with a ResNet-based encoder, followed by a decoder that incorporated atrous convolutions for upsampling [10,13,14]. This resulted in feature maps at higher sampling rates. Another enhancement is the so-called ‘Spatial pyramid Pooling’, where the idea is to pool the network together with atrous convolutions of different rates, such as 6, 12, and 18, since the rate controls the number of holes and the filter rates are chosen in the pattern of a pyramid to extract as much information as possible. However, the DeepLab family also fails to deal with low-contrast or semi-dark images. Finally, the last enhancement is achieved by using ML concepts or graphical probabilistic methods. A lot of methods were presented using super-pixel creation methods [18] for preprocessing or conditional random fields (CRFs) for post-processing [10,18,19,20]. The CRFs are used in a fully connected manner, which again imposes architectural constraints that require using same-sized images. However, they provide better object boundary localization with an additional network overhead. Furthermore, the super-pixel methods are used to pool the boundary information as an extra feature for the network, which increases the network overhead.

Table 1 presents a critical analysis of the closely related research that further supports the rationale and need for a generalized semantic segmentation solution for all scenarios (including semi-dark images).

The critical analysis presented in Table 1 indicates the need for DL mechanisms that hold for semi-dark images as well. The existing mechanisms solely focus on the design of deeper structures, which further increase the processing load, along with other complexities. Some of the solutions point to the usage of pre- and post-processing modules to sharpen the boundary pixels only, which is only one aspect of handling semi-dark images. Finally, only a few studies mentioned the occurrence of semi-dark images in the training and testing protocols. Thus, those methods cannot be generalized without further extended studies.

### 2.3. Critical Analysis of DeepLab Versions

DeepLab semantic segmentation networks are state-of-the-art models proposed by the Google research group. This model follows the increased-resolution feature-encoding-methods flow. The dense predictions are accomplished using atrous convolutions for the upsampling.

DeepLabV1

Chen et al. [10] proposed a semantic segmentation network using VGG-16 as the backbone network, and on top of the VGG feature map, probabilistic graphical methods, particularly conditional random fields (CRFs), are used. However, the network performance is not high enough since some failure modes were reported regarding dynamic light conditions and occlusions. Another study claimed that the proposed methods failed to integrate the features of CNN with CRF, limiting its overall functionality, though they could be efficiently integrated using architectural schemes presented in [19].

2.DeepLabV2

DeepLab network V2 incorporates spatial pyramid pooling into the previous architecture for better semantic segmentation accuracy. This version also replaces VGG-16 with ResNet [12] to tackle features with reduced spatial resolution caused by upsampling and downsampling; meanwhile, the atrous algorithm was used so that the DCNN works in the fully convolutional mode. The atrous algorithm upsamples the filters in subsequent convolutional layers, which results in feature maps at higher sampling rates by means of the insertion of holes between non-zero taps. The proposed version of the model still fails to capture the details of the object boundaries if they are sensitive to some of the results. DeepLabV2 uses atrous convolution in a post hoc manner; however, it is believed that integrating atrous convolution into the encoder would also affect the overall training by reducing the computational overhead. Moreover, the serial application of atrous convolution, as in [38], is expected to fetch accurate object semantic classes.

3.DeepLabV3

This version continues to use atrous convolution to provide a better receptive field. The enhancement is in terms of cascading the extra layers to include the global context by pooling the image-level features. The proposed network still fails to generate optimal results; as such, one aspect of necessary further investigation is performance analysis over different dilation rates. The study reported some of the misclassified results of the network where sofa/chair, dining table/chair, and the rear view of the objects were incorrectly classified in dynamic lighting conditions [13].

4.DeepLabV3+

The latest version of DeepLab ensures the fetching of rich contextual features, along with sharp object boundaries [14]. The network architecture still utilizes the concepts of encoder–decoder methods. The enhancement is in terms of bilinear upsampling of the encoder output, which is from DeepLabV3 in this case. The upsampling is performed using an upsampling factor value of 4 and then concatenating it with the low-level image features from the network base. The network performs better than the earlier versions; however, it still fails for some complex scenarios (congested with small objects of different scales), the rear views of objects, and for dynamic lighting conditions [14]. Moreover, the interpolation method can be further changed to bicubic interpolation to fetch most of the visual image information.

### 2.4. Conclusive Deductions

Indeed, every new version of DCNN has brought about better results in terms of accuracy. However, the enhancements focus on bringing deeper structures, which is acceptable if the results hold for all scenarios. However, many of the surveyed studies explicitly declared semi-dark images as a failure mode for these works. The improvements were made using unsupervised methods; however, the internal network problems still exist, such as slow training, covariate shifts, and the biased exploitation of the receptive field. These issues can be tackled by using appropriate atrous convolution and using appropriate normalization techniques within the DCNN as a part of a focused solution for semi-dark images. Inspired by the DeepLab architecture’s base rationale of controlling and neutralizing the exploitation throughout the image pixels (receptive field) by using atrous convolutions, Unified DeepLab was formulated. We proposed to extend this concept by presenting customized atrous rates, along with appropriate normalization techniques and non-linear connections to alleviate the effect of a covariate shift. Moreover, it is also recommended to use preprocessing for the creation of super-pixels so that local and global contexts are correlated to create the final segmentation mask.

### 2.5. Preliminary Hypothesis Validation

Based on the literary analysis, it was seen that most of the existing semantic segmentation solutions reported semi-dark images as the operational failure mode. Moreover, the conclusion of an extensive literature review resulted in hypothesis formulation based on our understanding of the problem at hand. It was hypothesized that lower image visibility increases the possibility of inaccurate semantic segmentation. To confirm this hypothesis, we trained the existing DeepLabV3+ version with the same data that was used for the benchmarking of the proposed solution and undertook related detailed qualitative analysis using DeepLabV3+ implementation in MATLAB. The results for semi-dark images were recorded using the addition and subtraction of offset values. These offset values served the purpose of providing image content visibility by increasing or decreasing the brightness. The examined images were processed to increase or decrease the brightness and then used for network training and testing, where all brightness value networks were independently trained and tested. The final performance results in terms of accuracy were then compiled and presented as graphical visualization. The final analysis results are shown in Figure 2.

Figure 2 shows that all the images with decreased brightness resulted in poor semantic segmentation results using DeepLabV3+. Up to an approximately 50–60% semantic segmentation accuracy was achieved for an offset value of minus 30 (Offset-30). However, the accuracy trend increased for all images with increased brightness. For an offset value of plus 30, the accuracy reached up to 80% for test case 6 (image: 001TP_007470), whose accuracy with a minus 30 offset was 60%.

Since the performance was substantially affected by the visibility of the image content, we further proposed a full-fledged framework to check the visibility of image content and then accordingly process the presented image for semantic segmentation. The framework was composed of three different layers, one of which identified the visibility of the image by using the RPLC (relative perceived luminance classification) algorithm [1], and the images were accordingly selected and propagated ahead for further processing. If RPLC classified any image as a semi-dark image, then it was passed to the proposed Unified DeepLab; otherwise, it was passed to the existing workflow based on DeepLabV3+.

## 3. Materials and Methods

### 3.1. Overview of Proposed Architecture

As stated in Section 2.3, the existing DCNNs used for the semantic segmentation provides constrained functionality for handling semi-dark images. These functionality limitations are attributed to several problems in the existing workflows. First, centric exploitations of the receptive field cause not all pixels of the image to contribute equally to the final calculated segmentation map. Second, the residual connections, particularly identity connections, result in an internal covariate shift, which results in a loss of representational power. Third, neighboring pixels have a higher probability of having the same class label due to holding closely related information. However, the classes of the pixels are still calculated independently based on per-pixel primitives. Finally, the backbone network can also increase or decrease the performance in terms of accurate segmentations and increased or decreased network loads. These problems in turn generate poor semantically segmented masks for semi-dark images. To solve these issues, the proposed framework consists of four components: a customized pyramid module for semi-dark images to reduce the effect of centric exploitations. The customized atrous convolution layers are selected in the form of a pyramid structure to extract as much information as possible without increasing the network’s processing load. Non-linear shortcuts are used for the encoder since the backbone network ResNet uses identity shortcuts, resulting in limited representational power. The identity shortcuts are replaced by non-linear ReLU connections and group normalization is used to reduce the effect of internal covariate shifts while training the network. A MobileNet encoder is used as a lightweight model to provide concise depth-wise separable convolutions. Local context segmentation is used to keep track of the local and global context of the image pixels by grouping neighborhood pixels based on certain criteria. Figure 3 illustrates the organization and flow of each component. In the first step, we used local context segmentation as a preprocessing layer for creating super-pixels, where the super-pixeled image used for the experiments was processed using 5000 super-pixels. In the second step, the locally grouped pixels were passed to the proposed parallel DCNN for semantic segmentation. Finally, statistical calculations were performed over the resultant map to create a high-definition accurate semantic segmentation map based on the parallel processing.

#### Rationale for Proposing an Ensemble Approach

The existing semantic segmentation solutions provide constrained functionality due to several intrinsic problems of convolutional neural networks. Therefore, in this research and based on the literary analysis, we first identified the problems that hinder the fetching of the optimal semantic class of an image pixel. Based on the identified problems, this research proposed an ensemble-based approach as a unified solution to tackle all the identified problems. Although the ensemble approach is believed to produce complexity, the proposed solution produces better semantic segmentation results for semi-dark images by keeping a good balance between complexity and accuracy. The introduction of non-linear shortcuts in the ResNet encoder was chosen to decrease the effect of a covariate shift arising during the training phase. Second, the MobileNet encoder was chosen to fetch smaller details of the visual scene due to the presence of greater network layers, which led to the attainment of richer image information. Finally, a customized decoder was proposed for the creation of image understanding at multiple spatial levels (lower to higher) by using customized atrous rates specifically for semi-dark images.

We further introduce the details of each component of the proposed framework in the following subsections.

### 3.2. Encoder Enhancement—Non-Linear Shortcuts

In encoder–decoder architectures, encoders play a crucial role in extracting the features from a visual image. One of the encoders that is widely used in the DeepLab architectures is ResNet. ResNet is known for its identity shortcuts in DCCNs. These shortcuts basically pool the information from the initial layers for the later layers to retain better spatial information of the extracted features. However, problems related to representational power persist. To solve this problem, non-linear shortcuts were proposed [15], along with a group normalization layer [17] after the non-linear ReLU connections. Figure 4 shows the existing ResNet connections, along with the updated ResNet layering arch, where the encoder was named RGSNet (ReLU-Group Norm ResNet).

To improve the performance, identity shortcuts were proposed for the DCCNs. Although the identity shortcuts improve the gradient stability, thus resulting in improved performance, the representational power is reduced due to the occurrence of covariate shifts during the network training. Due to this phenomenon, the network learns less during training. To find out the balance between representational power and gradient stability, along with the residual shortcuts, nonlinear group normalization shortcuts were introduced. The group normalization is better for non-linear shortcuts, as it does not require normalization along the batch direction. Thus, it gives an advantage for lightweight memory demands as opposed to batch normalization [39]. Following the architectural updates proposed in [15], a non-linear ReLU activation function plus group normalization is employed for the ResNet-based encoder, ultimately producing RGSNet. RGSNet normalizes the contribution of extra gradient stability in the presence of non-linear connections by using group normalization while involving minimal reengineering effort. The hyperparameter G (number of groups) was set to 16 in our experiments, as in [17]. This connection can be mathematically presented as
(1)$$y_{l}=h\left(x_{l}\right)+F\left(x_{l},W_{l}\right)\;\mathrm{and}\;x_{l\;+\;1}=f\left(y_{l}\right)$$

Here, in Equation (1), F represents the residual block and $W_{l}$ represents the learnable weights. $x_{l}$ and $x_{l\;+\;1}$ represent the inputs to layers l and l + 1, respectively, while h and f represent the identity mapping function of each layer, respectively. This follows the direct path of information propagation throughout the network.

### 3.3. MobileNetV2 Encoder

As the goal was to extract optimal information from the visual scene with high accuracy while keeping the mathematical operations as low as possible, this same concept was used for the implementation of the residual networks, such as ResNet. However, there exists another architecture that is based on the integration of inverted residual connections and ensures a lightweight nature for limited-memory applications. The problem with residual networks is that they work using the wide → narrow → wide approach, focusing the number of channels. These channels are compressed using a 1 × 1 convolution, followed by a 3 × 3 convolution with fewer parameters, and followed by a 1 × 1 convolution to again increase the number of channels. In contrast, MobileNet (specifically version 2) works using a narrow → wide → narrow approach to limit the number of parameters involved. MobileNetV2 accomplishes this task by widening the network using 1 × 1 a convolution and afterward uses a 3 × 3 depth-wise convolution to reduce the number of parameters involved. This entire scenario ensures accurate information extraction from semi-dark images, as the other components of the parallel DCNN are built using less deep network bases (ResNet18-depth18, no of parameters = 11.7 million). We used MobileNetV2 as the DeepLab encoder due to its lightweight nature with deeper and richer feature extraction; it also provides features to deal with linear bottlenecks (using ReLU6 rather than Simple ReLU) and inverted residual connections with a smaller number of parameters to be handled with only one additional hyperparameter t, which is the expansion rate of the channels and is set to a value of 6 (by default). For the semi-dark images, as the objective was to retrieve as much information as possible, this can be achieved with deeper networks. However, if we increase the number of layers in the network (such as by using ResNet), it increases the overhead, providing overall slow training and testing. For this reason, to benefit from the deep network and lightweight feature of MobileNet, it was employed to increase the detailed information of the visual scene. Eventually, it was found that incorporating MobileNetV2 with a depth of 53 (no. of parameters = 3.5 million) did not increase the memory footprint by much compared to the deeper networks, e.g., ResNet-50 with a depth of 50 (no. of parameters = 25.6 million) and ResNet-101 with a depth of 101 (no. of parameters = 44.6 million).

### 3.4. Decoder Enhancement—Customized Pyramid Module

The DeepLab network architecture family uses spatial pyramid pooling over the receptive field to mitigate the effect of centric exploitations. The existing pyramid pooling layers are created by inserting holes into the kernel to widen the receptive field; this phenomenon is called atrous convolutions [14]. The idea is to apply the atrous convolutions in parallel with different dilation rates, which controls the size of the filter. For the simple convolutions, a 3-by-3 convolution filter has nine parameters and a resultant receptive field of 3 by 3. However, when using dilated convolutions with a 3-by-3 convolution filter and a dilation rate of 2, the resultant receptive field turns out to be of shape 5 by 5, thus we obtain a larger perspective for the feature being calculated. By choosing the correct dilation rate and pooling a couple of atrous convolution layers, the effect of centric exploitations is mitigated. Finally, each feature map obtained from the atrous convolution is concatenated and propagated ahead in the network for the pixel classification. Figure 5 illustrates the customized decoder module for semi-dark images to optimally manipulate the pixel information.

Using the concept of atrous convolution, the updated kernel after the insertion of holes can be calculated using
(2)$$\hat{k}=k+\left(k-1\right)\left(d-1\right)$$

Equation (2) represents the calculation of the resultant filter/receptive field. $\hat{k}$ is the resultant filter, k is the actual filter size, and d is the dilation rate [40]. For the proposed decoder module, we used rate values of 3, 8, 13, 18, and 23 with receptive fields (updated kernels) of 7, 17, 27, 37, and 47, respectively. However, the recent DeepLab architecture uses dilation rates of 6, 12, and 18 with receptive fields (updated kernels) of 13, 25, and 37, respectively. These values are acquired by setting the existing filter size k equal to 3. From these numbers, it can be expected that the DeepLab loses pixel information due to the smaller receptive field, which was further demonstrated in the results given in Section 4.

### 3.5. Statistical Class-Wise Fusion

The statistical fusion performed on the network’s output is employed in such a way that it retains the final optimal class-wise segmentation map. For any DCNN, the objective function is given as Equation (3).
(3)$$J\left(\theta \right)=\log P\left(y|x;\theta \right)=\sum_{m=1}^{M}\log P\left(y_{m}|x;\theta \right)$$
where θ is the parameter vector for DCNN. The pixel label distributions are calculated using Equation (4).
(4)Pym|x;θ∝expfmym|x;θ
where fmym|x;θ represents the output of the DCNN at pixel m. Given this information, our statistical module follows the steps mentioned in Algorithm 1 for the proposed parallel DCNN for semi-dark images.
**Algorithm 1: Proposed Parallel DCNN for Semi-Dark Images****Input**: Initial DCNN parameters θ ∀ RGSNet-DeepLab (Net1), MobNet-DeepLab (Net2), Custom-DeepLab (Net3), customization parameters bl (including expansion rate, dilation rate, and group normalization number), segmented image Is with M pixels, pixel label set to y (1 → *n*, no of classes). **Parallel DCNN Processing Steps:**1: For each image pixel m, all three DCNN variants perform fml)=fm(l|x;θ+bl given yl=1−n.2: Every pixel (m → M, where M is the no. of image pixels) of image Is receives a semantic label yl from Net1, Net2, and Net3.**Statistical Class-wise Fusion Step:**3: Class-wise mapping is achieved using funi,m=maxfm1l,fm2l,fm3l
where fm1l is a class-wise label generated from Net1, fm2l is a class-wise label generated from Net2, and fm3l is a class-wise label generated from Net3.**Output:** Semantic segmentation pixel mask representing n object classes.

In particular, the final semantic segmentation masks resulting from each proposed network component are further analyzed to capture the optimal class-wise object prediction performance. The parallel DCNN component values are identified using the following algorithm (Algorithm 2).
**Algorithm 2: Parallel DCNN Class-Wise Fusion**1:Provided class-wise results of RGSNet ($f_{m1}\left(l\right)$), CustomDecoder-DeepLabV3+ ($f_{m2}\left(l\right))$, and MobileNet-DeepLabV3+ ($f_{m3}\left(l\right))$
2:Compare ($f_{m1}\left(l\right)$), ($f_{m2}\left(l\right))$, and ($f_{m3}\left(l\right))$
3:For each object class 4:     If ($f_{m1}\left(l\right)$) > ($f_{m2}\left(l\right)$) && ($f_{m1}\left(l\right)$) > ($f_{m3}\left(l\right)$)5:           Store $f_{m1}\left(l\right)$ in stack6:   Else If ($f_{m2}\left(l\right)$) > ($f_{m1}\left(l\right)$) && ($f_{m2}\left(l\right)$) > ($f_{m3}\left(l\right)$)7:           Store $f_{m2}\left(l\right)$ in stack8:     Else If ($f_{m3}\left(l\right)$) > ($f_{m1}\left(l\right)$) && ($f_{m3}\left(l\right)$) > ($f_{m2}\left(l\right)$)9:           Store $f_{m3}\left(l\right)$ in stack10:Create a final semantic segmentation vector by fetching the stack values

The final semantic segmentation map is the result of statistical class-wise fusion based on the maximum achieved value of the network performance.

The proposed system receives the preprocessed image in the form of locally grouped pixels via local context segmentation to keep track of the local pixel information in the form of super-pixels. All the DCNN hyperparameters were set to the default values, except the group normalization parameter and the filter dilation rates. The values of these parameters are the expansion rate t = 6, number of groups G = 16, and the dilation rates = 3, 8, 13, 18, and 23. Parallel DCNN processing steps are performed for the identification of each pixel label. After all the pixels obtain their class label, the results are fused to find the maximum of each class’s accuracy. Finally, the maximum of each class is fetched to generate the final semantic segmentation mask representing the optimal class labels for each pixel of the semi-dark image.

## 4. Experiments

In this section, we present the details of the dataset, implementation, evaluation criteria, and finally, the experimental analysis results.

### 4.1. Dataset and Evaluation Metrics

For the training and testing of the proposed Unified DeepLab version, we used the CamVid dataset [25,41]. This database was considered due to its complex nature, as it provides images of road settings where the dynamic lightning conditions are an intrinsic property. The dataset originally had 32 classes; however, for simplicity, we merged the class labels and undertook our analysis based on 11 classes (namely, sky, building, pole, road, pavement, tree, sign symbol, fence, car, pedestrian, and bicyclist). The merging was performed solely for the purpose of inducing simplicity in the overall experiments. However, the merging was performed based on the condition that none of the categories were incorrectly merged. The actual categories, along with the final merged categories, are shown in Table 2.

The classes/categories were merged based on visual relevance and similarity of the base classes, such as the bicyclist merged category being based on the bicyclist and motorcycle/scooter classes since both are two-wheeled objects. Therefore, to reduce the network training burden, similar categories were merged.

The dataset consisted of high-quality and high-resolution images of ego-motive scenes. The captured images were of resolution 960 × 720. Given that the emphasis of the presented research was the optimal semantic segmentation of semi-dark images, we used the concept of related perceived luminance classification presented in [1] to identify only the problem domain images. The resultant dataset contained 549 images. The filtered images were randomly divided into training, testing, and validation sets with proportions of 60:20:20, i.e., 329, 110, and 110 images for training, testing, and validation, respectively. The random selection of images was achieved so that none of the images was selected twice by using MATLAB’s ‘randperm’ method. Moreover, the images were selected by ensuring that the new random values were greater than 3, meaning that even if the random number turned out to be close to the previous number, it was rejected. The rejection of a random number was done to make sure none of the sequenced images were selected. The reason for not selecting the immediate sequenced image was to ensure that no similar image with a minor change was selected, as the data was coming from an image sequence generated from consecutive video frames. Moreover, if the same images were selected, then the network might not produce the desired result and end up with an overfitted model. However, as seen in Figure 7 for all the proposed parallel networks, the training (blue line) and the validation curves (black lines) were close to each other, implying that the network was not overfitted. Figure 1 shows a sample raw image, along with the ground truth mask and the respective class labels. Some of the sample images, particularly locally grouped super-pixeled images, along with the ground truth labels, are presented in Table 3.

#### 4.1.1. Proposal of Novel Evaluation Criteria

For the evaluation and benchmarking of the proposed solution, we report the overall accuracy (OA), intersection over union (IoU), and MeannBFScore.
(5)$$\mathrm{OA}=\frac{\mathrm{TP}+\mathrm{TN}}{\mathrm{TP}+\mathrm{FP}+\mathrm{TN}+\mathrm{FN}}$$
(6)$$\mathrm{IoU}=\frac{\mathrm{Area}\;\mathrm{of}\;\mathrm{Overlap}}{\mathrm{Area}\;\mathrm{of}\;\mathrm{Union}}$$
(7)$$\mathrm{Mean}\;\mathrm{BFScore}=\frac{\mathrm{Area}\;\mathrm{of}\;\mathrm{Overlap}}{\mathrm{Total}\;\mathrm{number}\;\mathrm{of}\;\mathrm{Pixels}}$$

In Equation (5), TP, FP, TN, and FN represent ‘True Positive’, ‘False Positive’, ‘True Negative’, and ‘False Negative’, respectively. These criteria do not suit the scenarios where there is a class imbalance since they produce unreliable results. Equations (6) and (7) are better representatives of the semantic segmentation problem. In Equation (6), the ‘area of overlap’ shows the overlap between the predicted segmentation and the ground truth, and the ‘area of union’ shows the union of the prediction and the ground truth. The values range from 0–100 (no overlap to perfectly overlapped prediction). In Equation (7), the ‘area of overlap’ is the same as in Equation (6), whereas the ‘total number of pixels’ represents the number of pixels present in the image.

##### Significance of Proposed Evaluation Criteria

The existing semantic segmentation evaluation criteria have both strengths and limitations. Many studies are presented in the literature regarding reasoning and applicability in all scenarios. The semantic segmentation score for problem-centric semantic class prediction was employed due to the questionable performance of a single evaluation metric, i.e., OA [42,43]. The unreliability of results is attributed to imbalanced datasets. However, a literature analysis showed that OA can be used in accordance with the other metrics in scenarios where the class imbalance is mitigated. To the best of our knowledge, there has not been any evaluation metric that takes into account the information of three widely used evaluation criteria to ensure the optimal retrieval of a pixel class. Thus, we present a new metric of evaluation for semantic segmentation, i.e., $S^{3}core$. This novel metric is a weighted combination of the OA, IoU, and Mean BFScore. Since it is a combination of three evaluation criteria, we named it ‘*S cube core*’ and it can be calculated using Equation (8).
(8)$$S^{3}core=W_{1}\times OA+W_{2}\times IoU+W_{3}\times Mean\;BFScore$$
where $W_{1}$, $W_{2}$, and $W_{3}$ are the weights associated with the metrics. For the weight analysis, we initially hypothesized randomly dividing the weights between the three evaluation criteria but this did not help much. However, then we expanded our experimental analysis by incrementing the weight of only one of the metrics and equally dividing the remaining weight among the other metrics. The experimental analysis of $S^{3}core$ is presented in Section 4.4, along with the recommended weights.

##### Experimental Protocol for Weight Identification of $S^{3}core$

The existing segmentation analysis was solely based on one of the above-mentioned evaluation criteria in Section 4.1.1. However, considering the need for accurate semantic class identification, generalized metric evaluation criteria were required. The overall accuracy (OA) metric failed to provide an accurate representation of the image pixel semantics in the presence of class imbalance. However, for the situations where the class imbalance is mitigated properly, it was found to be a useful analysis criterion. Meanwhile, the IoU and Mean BFScore remained better representations in terms of low false positives and false negatives. Considering these aspects, a novel semantic segmentation evaluation criterion, namely, $S^{3}core$, is presented. $S^{3}core$ takes into account all three of these existing evaluation criteria and generates a generalized weighted score. The obtained score value is applicable to all applications, where even small details of the visual scene are crucial for the final decision. The value of $S^{3}core$ is calculated using Equation (8).

For the weight optimization and analysis, different manipulation strategies were used for weight identification. A total of one thousand random weight numbers were analyzed for each of the selected strategies. The weights were selected in such a way that the sum of the weight values was always one for all the strategies. First, the strategy calculated the weights using incremental values of weights for the MeanBfScore and dividing the weights equally between the OA and IoU. The second and third strategies calculated the weights using incremental values of the IoU and OA, respectively, and assigned equal weights to the other two criteria in each case. The fourth strategy calculated randomized weights. The algorithms that were used for the identification of weights are presented in Algorithms 3–5. Another weight identification method employed was based on randomization. Here, again, the condition that the sum of the weights, i.e., ‘W’, should not exceed 1 was imposed and 1000 different random numbers were checked for the final optimized weight proposal. The weights in this strategy were identified using steps 4, 5, and 6 stated in Algorithm 6. For the random value generation, the Mersenne Twister algorithm (MT19937) [44] was used, which generates values greater than 0 and less than 1. Again, for each iteration, $S^{3}\;core$ values were computed.
**Algorithm 3: Incremental IoU**1:Initialize weights $W_{1}$ (weight for OA), $W_{2}$ (weight for IoU), $W_{3}$ (weight for MeanBFScore), and Overall_Weight W2:Set $W_{2}$ = 0.0013:Repeat until W2 == 14:Set $W_{1}$
$=(1-W_{2})/2$
5:Set $W_{3}$
$=W_{1}$
6: 
$W_{2}+=0.001$
7: 
$W=W_{1}+W_{2}+W_{3}$
8:If W > 19:  Break

Algorithm 3 presents the flow of the experiment for weight identification after initializing the variables. $‘W_{2}’$, which is the weight for the IoU, was set to 0.001 and was incrementally changed up until reaching a value of 1; in this way, the final set of iterations produced 1000 different weight values. Based on the value of $‘W_{2}’$, the weights $‘W_{1}’$ and $‘W_{3}’$ were identified for each iteration using steps 4 and 5. For each iteration, the weight sum ‘W’ was also checked such that the sum of all weights was always one. Moreover, for each of these weight values, the semantic segmentation performance metric (obtained using the DCNNs) was checked to calculate the final $S^{3}core$ value using Equation (8). The final semantic segmentation scores are presented in Table 9. A similar weight identification strategy was used for the incremental MeanBfScore and OA; for detailed steps, refer to Algorithms 4 and 5.
**Algorithm 4: Incremental MeanBFScore**1:Initialize weights $W_{1}$ (weight for OA), $W_{2}$ (weight for IoU), $W_{3}$ (weight for MeanBFScore), and Overall_Weight W2:Set $W_{3}\;$= 0.0013:Repeat until W3 == 14:Set $W_{1}$
$=(1-W_{3})/2$
5:Set $W_{2}$
$=W_{1}$
6:   $W_{3}+=0.001$
7:   $W=W_{1}+W_{2}+W_{3}$
8:If W > 19:    Break


**Algorithm 5: Incremental OA**
1:Initialize weights $W_{1}$ (weight for OA), $W_{2}$ (weight for IoU), $W_{3}$ (weight for MeanBFScore), and Overall_Weight W2:Set $W_{1}$ = 0.0013:Repeat until W1 == 14:Set $W_{2}$
$=(1-W_{1})/2$
5:Set $W_{3}$
$=W_{2}$
6:   $W_{1}+=0.001$
7:   $W=W_{1}+W_{2}+W_{3}$
8:If W > 19:    Break

Another weight identification that was employed was based on randomization. Here, again, the condition that the sum of the weights, i.e., ‘W’, should not exceed 1 was imposed and 1000 different random numbers were checked for the final optimized weight proposal. The weights in this strategy were identified using steps 4, 5, and 6 stated in Algorithm 6. For the random value generation, the Mersenne Twister algorithm (MT19937) [44] was used, which generates values greater than 0 and less than 1. Again, for each iteration, $S^{3}\mathrm{core}$ values were computed.
**Algorithm 6: Randomized Weights**1:Initialize weights $W_{1}$ (weight for OA), $W_{2}$ (weight for IoU), $W_{3}$ (weight for MeanBFScore), Overall_Weight W and counter C2:Set C = 13:Repeat until C <= 10004:   Set $W_{1}=0.3*\mathrm{Rand}()$
5:Set $W_{2}=0.3+0.3*\mathrm{Rand}()$
6:Set $W_{3}$ = $1-(W_{2}+W_{3}$)7:   C+=1
8:   $W=W_{1}+W_{2}+W_{3}$
9:If *W* > 110:    Break

The final proposed weights for $S^{3}\mathrm{core}$ were based on the analysis of four thousand different weight values. The final optimal weights were found via the incremental IoU weighting strategy, where the numerical analysis is presented in Table 9.

### 4.2. Implementation Details

#### 4.2.1. Experimental Setup

To conduct the experiments, we used a machine with the following specifications: Intel^®^ Core™ i7-10750 H CPU @2.60 GHz with 16.0 GB RAM on a 64-bit operating system powered using NVIDIA GeForce GTX. The code was written in MATLAB version R2021a, specifically using the Deep Network Designer toolbox for network design and conceptualization. To train the system, the parameter values were set as shown in Table 4.

#### 4.2.2. Handling Class Imbalance

Before training the network, we first checked the dataset statistics to identify whether any class imbalance existed. Since class imbalance can be detrimental to the training process due to unbiased behavior toward the dominant classes, it is imperative to identify and mitigate the effect of class imbalance. To determine the class imbalance, we counted the pixels by class labels. For this purpose, we used the countEachLabel method, which takes in a datastore (images) and returns pixel labels and a count for the input datastore; the result was named ‘frequency’, which is basically the average of the pixel counts. The pixel label visualization is shown in Figure 6, and it was seen that sky, building, road, and tree were the dominant classes, whereas pole, pavement, sign symbol, fence, car, pedestrian, and bicyclist had smaller numbers of class labels.

To balance the class labels, we used a weighting strategy. The weighting strategy was accomplished by identifying the median of the frequencies presented in Figure 4. The class weights were calculated using Equation (9).
(9)$$Class\;weight=median\left(frequency_{n}\right)./frequency_{n},\;n=1,2,\;\ldots \;,\;n$$

Here, n represents the number of classes and the class weight was identified using element-wise division, through which we obtained weights for each class. The final weights are given in Table 5.

Finally, the balanced class labels in terms of the weighting strategy were used for the network training.

#### 4.2.3. Training the Proposed Model Components

In this section, we report the training and validation performances that were obtained after setting the training parameters to the values specified in Table 2. For the benchmarking of the proposed Unified DeepLab, we trained ResNet-DeepLab, where different components of Unified DeepLab were trained separately and the accuracy and loss curves of each of the components are presented in Figure 7.

Figure 7a–c shows that the base ResNet-DeepLabV3+ reached the maximum validation accuracy after 16 epochs and completed 650 iterations. Similar results were observed with RGSNET-DeepLab and Customized Decoder-DeepLab provided that both the networks had extra layers and customization operators at the backend. Here, we could conclude that the memory footprint was not increased by enhancing the network architecture. Meanwhile, the peak performance (validation accuracy) in the training cycle of MobileNet-DeepLab was achieved after 21 epochs in 850 iterations, which was acceptable, as MobileNet is three times deeper than ResNet-18 (refer to Section 3.3 for details). Integrating these distinct network components, we further elaborate on the results of the proposed Unified DeepLab in upcoming sections.

##### Ablation Study

In this section, we report the individual component performance based on the class-wise OA, IoU, and Mean BFScore for the dataset images. The results presented in Table 5 are for the entire dataset used for the analysis. The RGSNet-based encoder successfully fetched the small details of the visual scene, as it achieved the maximum performances for the classes pedestrian, sign symbol, and pavement. MobileNet-ResNet achieved the maximum accuracies for only two classes, namely, sky and pole, which can be further optimized in later research. The Customized Decoder DeepLab variant produced the best performance by achieving the maximum accuracies for six classes, namely, building, road, tree, fence, car, and bicyclist. The promising performance of this customized decoded variant was attributed to the better extraction and generalized manipulation of receptive field pixels. Ultimately, all these DCNN network components created an optimal ‘Unified’ semantic segmentation solution for semi-dark images by correctly handling different aspects of the DCNN mechanics.

In Table 6, bold numeric values show the best class-wise performer for each network component. Moreover, the novel metric values were seen to boost the performance up to a margin of 10% compared with the singular metric Mean BFScore (such as the classes fence and car). Finally, the overall average for each network component was also seen to increase by using the proposed criteria. The ablation study showed that the best performer was the Customized Decoder DeepLab version due to its relevance with the correctly chosen rate values, which effectively increased the FOV for the feature extraction.

To validate the results visually, Figure 8 shows the image areas where the base ResNet-DeepLab misclassified the image pixels, whereas other variants perform better in terms of visual results.

The image portions highlighted with pink boxes over the tested image show that ResNet-DeepLabV3+ misclassified road (class) pixels as pavement, whereas our proposed variants classified the pixels accurately in the corresponding regions. Table 7 shows some more visual results of semantic segmentation over the tested images.

### 4.3. Comparison with the State-of-the-Art Method

We conducted a comparative analysis of the CamVid dataset on the base ResNet- DeepLabV3+ against the proposed DCNN, i.e., Unified DeepLab. Table 8 shows that Unified DeepLab performed better than ResNet-DeepLabV3+ in terms of greater class-wise accuracy for up to seven classes out of the eleven predefined classes. Even for a sleek object, such as a bicyclist, a greater MeanBfScore was achieved with a margin of 4%. Overall, in all the other classes, the proposed network exceeded the existing network’s performance with a margin of 3%. Finally, the averaged class-wise values showed that the proposed network performed better than the existing network with margins of 3% for the overall accuracy (OA), approximately 1% for the IoU, and 2% for the Mean BFScore.

Table 8 also shows the analysis results of the existing evaluation metrics versus the proposed criteria. Again, a similar performance boost was witnessed for the state-of-the-art semantic segmentation solution and the proposed Unified DeepLab.

### 4.4. Semantic Segmentation Analysis Based on $S^{3}core$


The segmentation analysis was solely based on each of the above-mentioned evaluation criteria. However, considering the need for accurate semantic class identification, generalized metric evaluation criteria are required. The overall accuracy (OA) metric fails to accurately represent the image pixel semantics in the presence of class imbalance, though for the situations where the class imbalance is mitigated properly, it can be a useful analysis criterion. However, the IoU and Mean BFScore remain the better representations in terms of low false positives and false negatives. Considering these aspects, we present a novel semantic segmentation evaluation criterion, namely, $S^{3}core$. $S^{3}core$ takes into account all three of these existing evaluation criteria and generates a generalized weighted score. The obtained score value is applicable to all situations where even small details of the visual scene are crucial for the final decision. The value of $S^{3}core$ is calculated using Equation (8). For the weight optimization and analysis, we conducted experiments using different manipulation strategies for weight identification. A total of one thousand weight numbers were analyzed for each selected strategy. The weights were selected in such a way that the sum of the weight values was always one for each strategy. The first strategy calculated weights using incremental values of weights for MeanBfScore and divided the remaining weight equally between the OA and IoU. The second and third strategies calculated weights using incremental values of the IoU and OA, respectively, and assigned equal weights for the other two criteria in each case. The fourth strategy calculated randomized weights. The final weight analysis is presented in Table 9.

From Table 9, it can be seen that the existing accuracies could be integrated and the accuracy margins can likely be boosted by up to 10% if the correct weighting scheme is employed. The maximum $S^{3}core$ value was attained for the incremental weighting OA scheme, i.e., 85% for Unified DeepLab and 82% for ResNet-DeepLabV3+. However, based on the literary survey conducted, we recommend the usage of an incremental IoU weighting scheme as the IoU is a better representative of the semantic segmentation pixel-wise classification, but it still incorporates the information of the OA. Hence, we propose an incremental IoU weighting scheme, which led to boosted margins of up to 10% with the weight values $W_{1}$ = 0.4995, $W_{2}$ = 0.001, and $W_{3}$ = 0.4995 for the OA, IoU, and MeanBFscore, respectively.

## 5. Discussion

In this paper, we present a unified approach to solving different problematic aspects of DCNN fundamentals. Since semantic segmentation applications are not confined to only the autonomous vehicle industry, our proposed solution holds for all the related applications. The comprehensive analysis was based on high-resolution images of dynamic visual scenes that incorporated different complexities (including different-sized objects and relatively low-luminance images). Incorporating all these complexities, the proposed DCNN still managed to produce a high performance margin. The optimal performance requisition pointed to the applicability of this solution to other problem domain areas, as well where image complexities are supposed to be an intrinsic feature.

### Novelty and Contributions

In our considered opinion, the novelty and contribution of this research are of moderate nature, as highlighted below.

Novel ensemble model approach

The focused solution base incorporated different network architectural changes that were otherwise used for intelligent visual decisions (simple object detection) but not specifically semantic segmentation problems. The proposed ensemble was based on a modified model that was backed by some existing architectures. To the best of our knowledge, this model does not exist in the current literature. Moreover, the proposed solution is expected to benefit from various automated domain applications that require optimal information extraction for all scenarios (irrespective of object scale, color, and luminance scenarios).

2.Novel semantic segmentation evaluation criterion

A semantic segmentation criterion was proposed as a representative criterion for all the scenarios where data imbalance strategies are employed. Moreover, the higher values of $S^{3}core$ implied a good ranking of semantic segmentation performance against the ground truth labels and perceptual analysis. This perceptual analysis can be further comprehensively investigated by a subjective user study. The criterion is generic in nature and can be used by other researchers in the domain in either its current form or as a basis to create an appropriate extension to suit the needs of semantic segmentation applications.

## 6. Conclusions

We provided a unified approach based on statistical class-wise fusion, bringing together different problem solutions to facilitate optimal pixel class identification in semi-dark images. We improved the existing solution with minimal DCNN architectural changes. The four key changes made involved the usage of the following:(1)Preprocessed super-pixeled images, which are locally grouped pixel images that keep track of the local contexts of images;(2)Non-linear shortcuts followed by group normalization layers in a residual network encoder (ResNet-18) to increase the feature representational power and normalization to ensure improved training accuracy;(3)A MobileNet-based encoder to increase the depth of the network for fetching fine-grained details using a deeper but less complex network that involves 8.2 million fewer parameters compared with the base ResNet-18 encoder;(4)A customized pyramid decoder (customized dilated convolution layers) to provide focused control of the receptive field and mitigate the effect of centric exploitations in semi-dark images.

All these changes in an encoder–decoder-based DCNN provided a unified perspective that allowed for a generalized proposed solution, i.e., Unified DeepLab. The proposed Unified DeepLab was evaluated on the CamVid dataset and was benchmarked against the most recent and effective state-of-the-art counterpart, namely, DeepLabV3+. Our experimental analysis showed that the proposed DCNN outperformed DeepLabV3+ by a margin of 3% for the overall accuracy and 2% for the Mean BFScore. Finally, we also presented a novel evaluation criterion for a semantic segmentation score called ‘$S^{3}core$’ that was based on the weighted combination of the overall accuracy, IoU, and Mean BFScore, which further boosted the class-wise pixel performance by 10%.

## Figures and Tables

**Figure 1 sensors-22-05312-f001:**
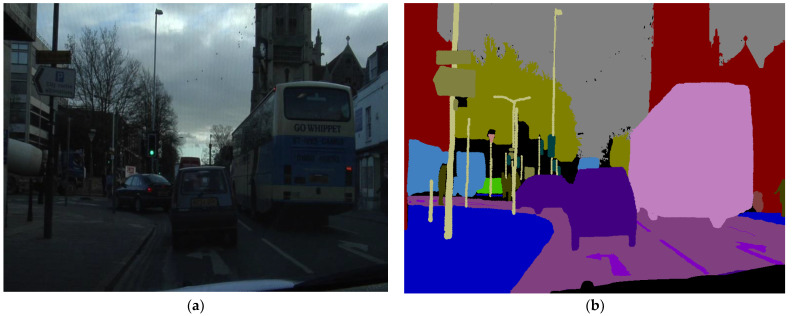
Semantic Segmentation Demonstration (**a**) Actual Image (**b**) Semantic Segmentation Ground-truth.

**Figure 2 sensors-22-05312-f002:**
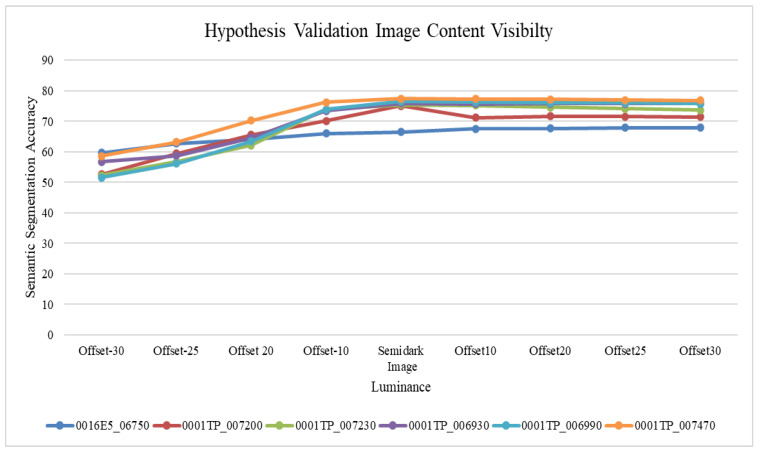
Hypothesis Validation.

**Figure 3 sensors-22-05312-f003:**
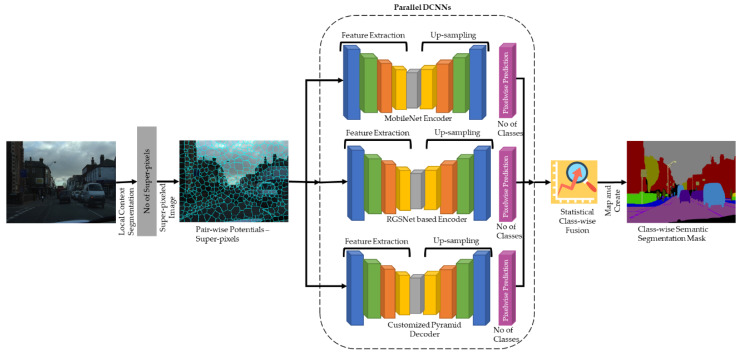
Proposed Parallel DCNN for Semi-dark Images.

**Figure 4 sensors-22-05312-f004:**
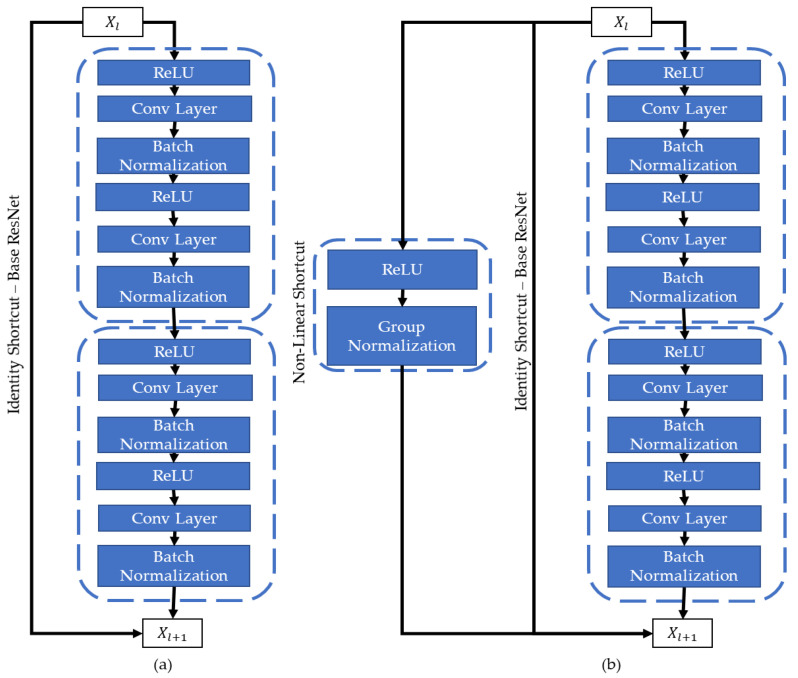
(**a**) ResNet with Identity shortcuts. (**b**) RGSNet (ReLU-Group Norm ResNet) with non-linear connections.

**Figure 5 sensors-22-05312-f005:**
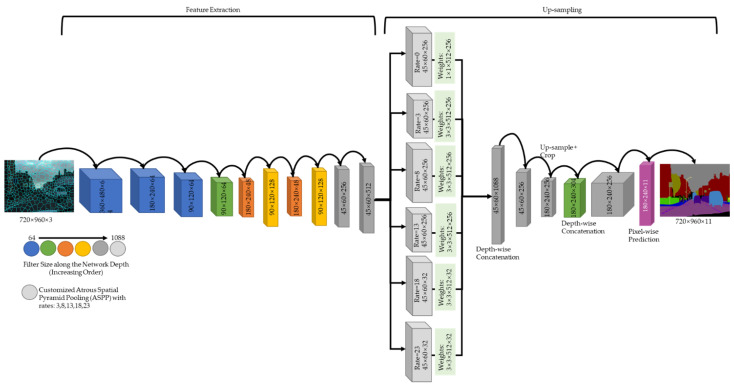
Customized Decoder Module for Semi-dark Images.

**Figure 6 sensors-22-05312-f006:**
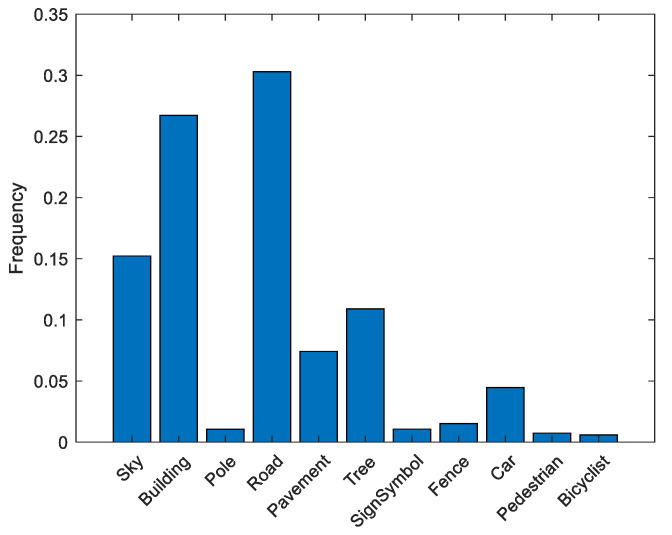
Class Distribution in the CamVid Dataset (Semi-dark Images).

**Figure 7 sensors-22-05312-f007:**
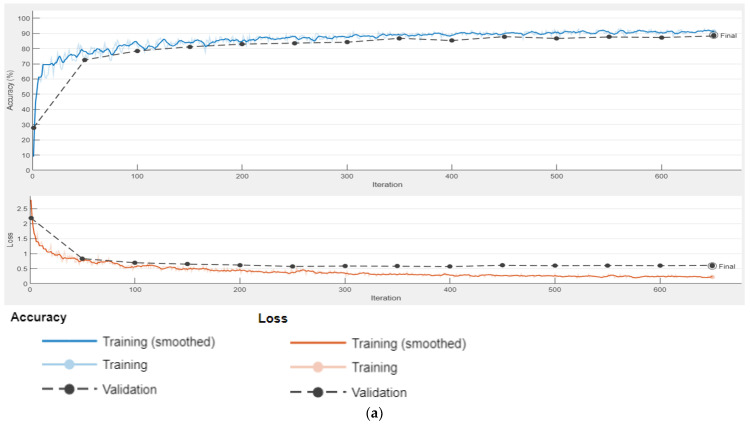
DCNN Accuracy and Loss Graphs. (**a**) ResNet-DeepLabV3+, (**b**) RGSNet-DeepLabV3+, (**c**) Customized Decoder-DeepLabV3+, (**d**) MobileNet-DeepLabV3+.

**Figure 8 sensors-22-05312-f008:**
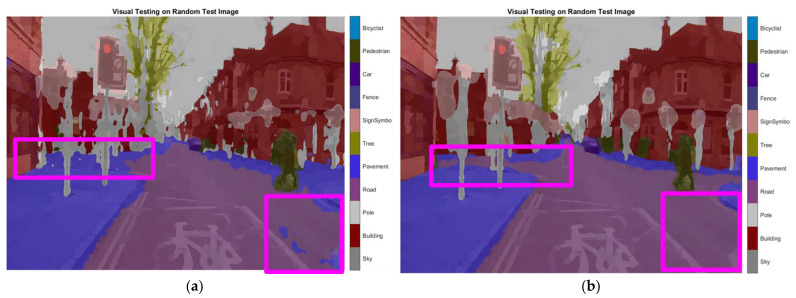
Visual Analysis of a Tested Image. (**a**) ResNet-DeepLabV3+, (**b**) RGSNet-DeepLab, (**c**) MobileNet-DeepLab, (**d**) Customized Decoder-DeepLab.

**Table 1 sensors-22-05312-t001:** Comparative Analysis of Semi-dark Image-Centric Semantic Segmentation.

Deep Learning Architecture	Original Architecture	Testing Benchmark	Semi-Dark Image Handling	Observations
VGG [9]	AlexNet [21]	ILSVRC-2012 Pascal VOC [22] MS Coco [23]	🗙	Poor localization due to the final DCNN layer [10] Does not use any unsupervised pre-training scheme to aid performance [21]
ResNet [12]	VGG [9]	ILSVRC-2012	🗙	Invariance to spatial transformations results in the loss of fine details [10] Limited representational power due to the usage of identity shortcuts [15,17]
		Pascal VOC [22] MS Coco [23]		
DeepLab [10]	VGG [9]	Pascal VOC [22] Cityscapes [24]	🗙	Fails to extract fine-grained details in semi-dark scenarios [10]
SegNet [11]	VGG [9]	Cam Vid Dataset [25] Sun RGB-D [26]	✓	Fails to capture pixel information and results in coarse segmentation results. Peak performance is achieved in the presence of handcrafted features [27]
DeepLabV2 [10]	VGG [9]	Pascal VOC [22] MS Coco [23] Cityscapes [24]	🗙	Better accuracy than the previous version but still cannot handle semi-dark images No preprocessing module is used for highlighting boundaries
DeepLabV3 [13]	DeepLabV2 [10]	Pascal VOC [22] Cityscapes [24]	🗙	Several different rates need to be further investigated, as the system still fails to handle complex images
DeepLabV3+ [14]	DeepLabV3 [13], ResNet [12]	Pascal VOC [22] Cityscapes [24]	🗙	Attains better accuracy than the previous version; however, it still fails to work for darker regions, complex images (congested with small objects of different scales), and the rear views of objects [14]
Attention Model [6]	Super-pixel creation (SLIC) DeepLab [10]	Pascal VOC [22] MS Coco [23]	🗙	Trained and tested to generate a person’s body parts, and fails to segment in scenarios of difficult human poses [6] Uses a preprocessing module for highlighting boundaries
Super-Pixel and CRF-Based FCN [18]		Cityscapes [24]	🗙	Uses CRFs as post-processing to aid the performance accuracy [18] Presents promising research results that can be extended to semi-dark images
Super-Pixel-Based Hierarchical Network [28]	Super-pixel creation (tree/clustering) ConvNet (2-layer MLP) CRF	SIFT Flow dataset [29] Barcelona dataset [30] Stanford background [31]	🗙	Focuses on the extraction of small details Pools the network with super-pixel boundary information
Semi-Supervised Convolution Neural Network [32]	DeepLab-CRF [10]	Pascal VOC [22] MS Coco [23]	🗙	Incorporates super-pixel information to provide better results No mention of performance for semi-dark images
Higher-Order CRF in DNN	FCN VGG [9]	Pascal VOC [22]	🗙	Fast RCNN for object detection Parallel implementation of an object-detection CNN with CRF and a graph-based method for super-pixel extraction can increase the complexity
Super-Pixel-Based DCNN for Road Segmentation [33]	VGG [9], ResNet [12]	Cityscapes [24]	🗙	Trained and tested in controlled road environments No mention of semi-dark images Uses both ResNet and VGG, which increases the memory consumption
Super-Pixel and Statistically Learned DCNN	Simple fully convolution neural network	Pascal VOC [22] Sun RGB-D [26]	✓	Super-pixels are pooled in the network as extra information
Spatio-Temporal Data-Driven Pooling DCNN [34]	CNN (No specifics written)	Sun RGB-D [26]	✓	Pro: spatio-temporal data-driven pooling can receive multiple images and their correspondence as input Uses prior super-pixels Unknown network base raises questions about the generic applicability
Super-Pixel- and CRF-Based DCNN [35]	SEGNET-VGG16 [11], DeepLabV2-ResNet [10]	LFW-PL dataset [36], HELEN dataset [37] (facial images) Pascal VOC [22]	🗙	Complex architecture with different base components

🗙→Absence of Semi-dark image handling, **✓**→ Presence of Semi-dark image Handling.

**Table 2 sensors-22-05312-t002:** Actual CamVid Classes vs. Merged Classes.

S#	Merged Class	Actual CamVid Dataset Class
1	Sky	Sky
2	Building	Bridge Building Wall Tunnel Archway
3	Pole	Column_Pole TrafficCone
4	Road	Road LaneMkgsDriv LaneMkgsNonDriv
5	Pavement	Sidewalk ParkingBlock RoadShoulder
6	Tree	Tree VegetationMisc
7	Sign Symbol	SignSymbol Misc_Text TrafficLight
8	Fence	Fence
9	Car	Car SUVPickupTruck Truck_Bus Train OtherMoving
10	Pedestrian	Pedestrian Child CartLuggagePram Animal
11	Bicyclist	Bicyclist MotorcycleScooter

**Table 3 sensors-22-05312-t003:** Super-pixeled Images and ground truth masks in the CamVid Dataset.

	Sample 1	Sample 2	Sample 3	Sample 4	Sample 5
Images	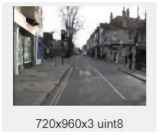	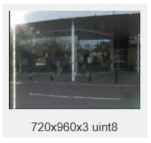	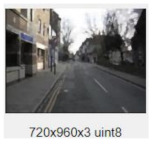	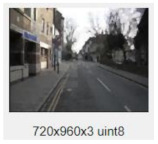	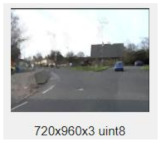
Ground truth	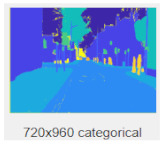	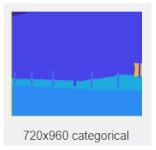	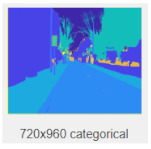	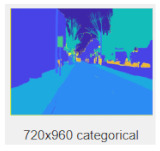	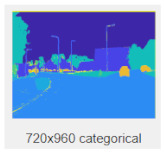

**Table 4 sensors-22-05312-t004:** Network Training Parameters.

Parameter	Solver	Initial Learning Rate	Validation Frequency	Maximum Epochs	Mini-Batch Size	L2 Regularization	Gradient Threshold Method	Validation Patience	Shuffle
Value	Sgdm (Stochastic Gradient Descent with Momentum)	0.01	50	25	8	0.0001	L2 Norm	5	Every-epoch

**Table 5 sensors-22-05312-t005:** Class Weights for Handling Biased Network Training.

Class	Weight Value
Sky	0.2933
Building	0.1674
Pole	4.2693
Road	0.1480
Pavement	0.5942
Tree	0.3766
Sign Symbol	4.0753
Fence	1.5294
Car	1.0000
Pedestrian	5.6283
Bicyclist	4.2795

**Table 6 sensors-22-05312-t006:** Class-wise Accuracy Analysis.

Network		Class	Sky	Building	Pole	Road	Pavement	Tree	Sign Symbol	Fence	Car	Pedestrian	Bicyclist	Average
	Metrics													
RGSNet-DeepLab	OA		0.9408	0.7491	0.5761	0.9241	**0.9028**	0.8509	**0.8152**	0.7119	0.8612	**0.8506**	0.8001	0.8166
	IoU		0.9046	0.7223	0.1657	0.9161	**0.6983**	0.7502	**0.2220**	0.5180	0.7009	**0.2914**	0.5184	0.5825
	Mean BFScore		0.8723	0.5457	0.4613	0.7616	**0.6879**	0.6343	**0.3308**	0.4337	0.6020	**0.4452**	0.4675	0.5675
	$S^{3}core$		0.9066	0.6475	0.5183	0.8429	**0.7952**	0.7426	**0.5726**	0.57278	0.7316	**0.6475**	0.6336	0.6919
MobileNet-DeepLab	OA		**0.9640**	0.7621	**0.6752**	0.9516	0.8901	0.8523	0.7258	0.7337	0.8819	0.7703	0.7852	0.8175
	IoU		**0.9128**	0.7343	**0.1605**	0.9378	0.7405	0.7474	0.3064	0.5638	0.7315	0.3551	0.5997	0.6173
	Mean BFScore		**0.9015**	0.5374	**0.4609**	0.8305	0.7510	0.6375	0.4291	0.5046	0.6489	0.5114	0.5943	0.6188
	$S^{3}core$		**0.9327**	0.6498	**0.5676**	0.8911	0.8205	0.7449	0.5772	0.6191	0.7653	0.6405	0.6896	0.7180
Customized Decoder-DeepLab	OA		0.9498	**0.8438**	0.6427	**0.9519**	0.8695	**0.8899**	0.6828	**0.7604**	**0.8942**	0.7739	**0.8127**	**0.8247**
	IoU		0.9124	**0.8047**	0.2115	**0.9383**	0.7484	**0.7760**	0.3663	**0.6091**	**0.7558**	0.4198	**0.5922**	**0.6486**
	Mean BFScore		0.8913	**0.6405**	0.5579	**0.8295**	0.7706	**0.6858**	0.4914	**0.5272**	**0.6834**	0.6007	**0.6179**	**0.6633**
	$S^{3}\mathrm{core}$		0.9205	**0.7422**	0.5999	**0.8907**	0.8199	**0.7878**	0.5868	**0.6437**	**0.7887**	0.6870	**0.7151**	**0.7438**

**Table 7 sensors-22-05312-t007:** Visual Semantic Segmentation Results.

Test Case	Network			
	ResNet-DeepLabV3+	RGSNet-DeepLab	MobileNet-DeepLab	Customized Decoder-DeepLab
1	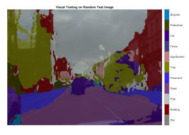	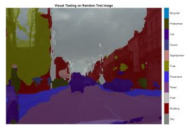	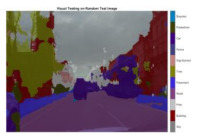	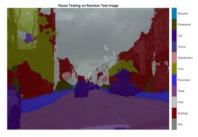
2	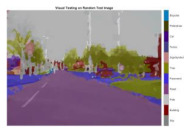	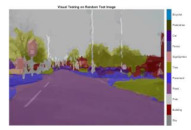	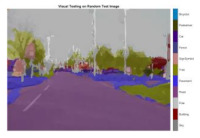	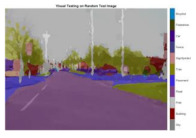
3	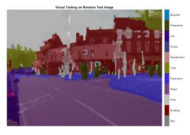	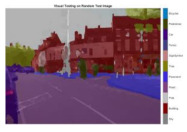	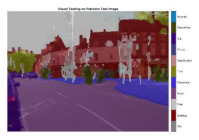	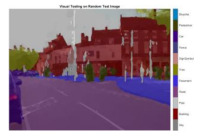
4	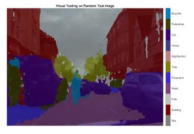	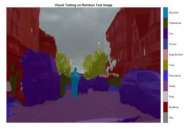	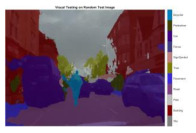	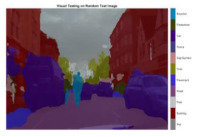

**Table 8 sensors-22-05312-t008:** Class-wise Comparative Analysis with the State-of-the-Art DCNN.

Network		Class	Sky	Building	Pole	Road	Pavement	Tree	Sign Symbol	Fence	Car	Pedestrian	Bicyclist	Average
	Metrics													
ResNet-DeepLabV3+	OA		0.9486	0.8418	0.6557	0.9536	0.8779	0.8533	0.6782	0.7724	0.8812	0.8064	0.8261	0.8268
	IoU		0.9100	0.7976	0.1938	0.9406	0.7536	0.7625	0.399	0.5736	0.7402	0.3970	0.5937	0.6420
	Mean BFScore		0.8942	0.6253	0.5299	0.8348	0.7779	0.6674	0.5191	0.4978	0.6567	0.5621	0.5778	0.6494
	$S^{3}\mathrm{core}$		0.9214	0.7336	0.5923	0.8942	0.8278	0.7603	0.5984	0.6350	0.7689	0.6839	0.7018	0.7380
Unified DeepLab	OA		**0.9640**	**0.8438**	**0.6752**	0.9519	0.9028	**0.8899**	0.8152	**0.7604**	**0.8942**	**0.8506**	**0.8127**	**0.8510**
	IoU		**0.9128**	**0.8047**	**0.2115**	0.9383	0.7484	**0.7760**	0.3663	**0.6091**	**0.7558**	**0.4198**	**0.5997**	**0.6493**
	Mean BFScore		**0.9015**	**0.6405**	**0.5579**	0.8305	0.7706	**0.6858**	0.4914	**0.5272**	**0.6834**	**0.6007**	**0.6179**	**0.6643**
	$S^{3}\mathrm{core}$		**0.9327**	**0.7422**	**0.6161**	0.8912	0.8366	**0.7878**	0.6530	**0.6437**	**0.7887**	**0.7253**	**0.7151**	**0.7575**

**Table 9 sensors-22-05312-t009:** Comparative Analysis with $S^{3}core$ (different weighting schemes).

Network	$S^{3}core\;\mathrm{Weighting}\;\mathrm{Scheme}$	$\mathrm{Min}\;S^{3}core\;(\mathrm{with}\;\mathrm{Weights})$	$\mathrm{Max}\;S^{3}core\;(\mathrm{with}\;\mathrm{Weights})$
ResNet-DeepLabV3+	Incremental MeanBFScore	64 $W_{1}$ = 0, $W_{2}$ = 0, $W_{3}$ = 1	72.99 $W_{1}$ = 0.4995, $W_{2}$ = 0.4995, $W_{3}$ = 0.001
Unified DeepLab	Incremental MeanBFScore	66 $W_{1}$ = 0, $W_{2}$ = 0, $W_{3}$ = 1	74.49 $W_{1}$ = 0.4995, $W_{2}$ = 0.4995, $W_{3}$ = 0.001
ResNet-DeepLabV3+	Incremental IoU	64 $W_{1}$ = 0, $W_{2}$ = 1, $W_{3}$ = 0	72.99 *W1* = 0.4995, $W_{2}$ = 0.001, $W_{3}$ = 0.4995
Unified DeepLab	Incremental IoU	64 $W_{1}$ = 0, $W_{2}$ = 1, $W_{3}$ = 0	75.48 $W_{1}$ = 0.4995, $W_{2}$ = 0.001, $W_{3}$ = 0.4995
ResNet-DeepLabV3+	Incremental OA	64.02 $W_{1}$ = 0.0010, $W_{2}$ = 0.4995, $W_{3}$ = 0.4995	82 $W_{1}$ = 1, $W_{2}$ = 0, $W_{3}$ = 0
Unified DeepLab	Incremental OA	65.02 $W_{1}$ = 0.0010, $W_{2}$ = 0.4995, $W_{3}$ = 0.4995	85 $W_{1}$ = 1, $W_{2}$ = 0, $W_{3}$ = 0
ResNet-DeepLabV3+	Randomized	64.00 $W_{1}$ = 0.00008, $W_{2}$ = 0.48808, $W_{3}$ = 0.51184	69.38 $W_{1}$ = 0.29931, $W_{2}$ = 0.51779, $W_{3}$ = 0.18290
Unified DeepLab	Randomized	64.97 $W_{1}$ = 0.00863, $W_{2}$ = 0.59849, $W_{3}$ = 0.39288	71.02 $W_{1}$ = 0.2993822, $W_{2}$ = 0.33354, $W_{3}$ = 0.36708

## Data Availability

The following Camvid dataset was used: http://web4.cs.ucl.ac.uk/staff/g.brostow/MotionSegRecData/ (accessed on 29 May 2022).

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
