# Peer review of "Unified DeepLabV3+ for Semi-Dark Image Semantic Segmentation"

_sensors, 2022, doi:10.3390/s22145312_

Round 1

Reviewer 1 Report

In this paper, a module made up of three types of sub-networks is developed to handle the problem of semi-dark image segmentation. However, the authors were unable to present sufficient reasons for employing three sub-networks.  The module is further complex by three sub-networks. Furthermore, I have my doubts about whether the proposed evaluation method works because the paper does not provide adequate justifications for selected weight values.

Other problems:

1) Images used on the paper are too blurry.

2) The reference format has a problem that has to be addressed.

Author Response

Respected Reviewer, 

Thanks for the constructive review. We have substantially increased the quality of the manuscript based on your reviews. Kindly check the changes, we hope to hear from you soon. 

 Thanks

Reviewer 2 Report

This paper tried to address the issue of semantic segmentation of dark images using a proposed framework with three Deeplab v3 variations. Though the approach is technically sound, the experiments seem to be weak with such a small dataset. Please see the comments below:

1.     It may not be necessary to give such a detailed explanation of what semantic segmentation is (lines 144 – 175), since it should be commonly known by the readers.

2.     The reviewer appreciates the detailed critical analysis of existing works in Table 1.

3.     It seems like the proposed parallel DCNN shown in Figure 2 is a standard model ensembling process with merging the labels at the end. Is there any domain knowledge about the semi-dark image used in the fusion of labels at the end?

4.     Line 335, please elaborate on why MobilenetV2 is particularly useful for semi-dark images.

5.     Line 399, Please provide more detailed reasons on why the classes were merged during the experiments, is it related to the class imbalance issues discussed in line 455?

6.     It seems like the datasets used in this paper are very small (i.e., only 548 images), and the images were collected in sequence with a large overlap between each other. The random selection of training and testing sets (line 407) may cause the issue of overfitting the network which increases its performance since one image in the testing set may have a large overlap with the image in the training set, was this considered during the study?

7.     The experiments showed a performance improvement on the selected 548 images dataset. The reviewer is wondering how is the proposed method perform on the dataset with normal lighting? And what about its performance on a dataset that contains both dark images and images with normal lighting?

Author Response

(The authors gave the same response as above.)

Round 2

Reviewer 1 Report

The innovation of this article is relatively low, and it is just a combination of several mature technologies. Moreover, I also doubt the effectiveness of the combination. The only thing that is original is the so-called S_3 evaluation metric. This metric is actually the weighted sum of several common methods. The value of the weight is only obtained through experiments, lacks theoretical basis, and does not have generalization. The overall model experiment lacks comparison with other more popular semantic segmentation models.

Author Response

Respected Reviewer,

Thank you for the meticulous comments and let us improve the quality of the article. We have addressed your comments. Hoping to hear from you soon.

Reviewer 2 Report

The reviewer thanks the authors' effort in addressing the comments. Please make sure the errors shown on lines 575, 576, and 582 are corrected and references are accurate.

Author Response

(The authors gave the same response as above.)

Round 3

Reviewer 1 Report

I have no other opinion, and agree to accept

This manuscript is a resubmission of an earlier submission. The following is a list of the peer review reports and author responses from that submission.